# GRP78 Activity Moderation as a Therapeutic Treatment against Obesity

**DOI:** 10.3390/ijerph192315965

**Published:** 2022-11-30

**Authors:** Dongjin Pan, Yunzhu Yang, Aihua Nong, Zhenzhou Tang, Qing X. Li

**Affiliations:** 1Institute of Marine Drugs, Guangxi University of Chinese Medicine, Nanning 530200, China; 2Department of Molecular Biosciences and Bioengineering, University of Hawaii at Manoa, 1955 East-West Road, Honolulu, HI 96822, USA

**Keywords:** GRP78, molecular target, action mechanism, metabolic disorder, obese

## Abstract

Glucose-regulated protein 78 (GRP78), a molecular chaperone, is overexpressed in patients suffering from obesity, fatty liver, hyperlipidemia and diabetes. GRP78, therefore, can be not only a biomarker to predict the progression and prognosis of obesity and metabolic diseases but also a potential therapeutic target for anti-obesity treatment. In this paper, GRP78 inhibitors targeting its ATPase domain have been reviewed. Small molecules and proteins that directly bind GRP78 have been described. Putative mechanisms of GRP78 in regulating lipid metabolism were also summarized so as to investigate the role of GRP78 in obesity and other related diseases and provide a theoretical basis for the development and design of anti-obesity drugs targeting GRP78.

## 1. Introduction

Obesity is listed as a “first-class disease” with greater harm than infectious diseases in the World Health Organization and is mainly caused by energy intake and consumption imbalance, which can induce chronic metabolic diseases such as cancer, cardiovascular disease, diabetes, and hyperlipidemia [1]. The global annual treatment cost of obesity-related diseases is more than 10 billion U.S. dollars [2,3]. The research and development of anti-obesity drugs amount to several billion U.S. dollars [4]. At present, only two drugs (orlistat and sibutramine) have been approved by the U.S. Food and Drug Administration (FDA) for long-term treatment and to improve clinical obesity. Orlistat is the only over-the-counter (OTC) weight loss drug approved by China’s National Medical Products Administration. Orlistat is a hydrated derivative of lipstatin, which lowers body weight via inhibiting gastrointestinal lipase and reducing the absorption of food source fat. At the same time, sibutramine increases physiological hypersatiety and decreases appetite by inhibiting the reuptake of neurotransmitters such as serotonin, norepinephrine and dopamine. In addition, traditional Chinese medicines are commonly used, which include cassia seed, fleece-flower root, cattail pollen, hawthorn, rhubarb, gingko leaf, etc. The main active anti-obesity ingredients of traditional Chinese medicines were flavonoids and anthraquinone compounds [5], such as puerarin, curcumin, gallic acid, polyphenol, berberine, capsaicin, and emodin [6]. The Traditional Chinese Medicine Systems Pharmacology (TCMSP) database listed 25 anti-obesity targets, 467 herbs and 1592 relative ingredients (TCMSP database available online: http://tcmspw.com/tcmsp.php; accessed on 11 November 2022). However, due to the complicated causes and pathogenesis of obesity, there is no drug that can completely cure obesity at present.

What the cause of obesity is and how to balance energy intake and consumption at the cellular level have been unsolved scientific problems. The current hypotheses of obesity development mainly include the reactive oxygen species (ROS) theory, endoplasmic reticulum stress (ERS) hypothesis, energy balance hypothesis, microbial hypothesis, and thrifty gene hypothesis. For example, ERS can lead to mitophagy [7] and further lead to obesity [8]. Furthermore, the overload of energy intake can further aggravate the ERS activated by ROS after mitophagy.

The 78 kDa glucose-regulated protein (GRP78) is an important molecular chaperone, and its up-expression indicates ERS occurrence. Both clinical and animal studies have found that GRP78 overexpression is strongly correlated with obesity, and GRP78 is an essential component of ERS to balance mitochondrial biosynthesis and mitophagy, which is a potential therapeutic target for obesity [9]. However, the downstream mechanisms and pathogenesis of obesity mediated by GRP78, or traditional Chinese medicine, plays weight-loss and lipid-lowering roles via targeting GRP78 are still unclear.

A literature search with the keywords “GRP78” or “BiP/HSPA5” gave more than 20,000 publications in the Web of Science, PubMed, ACS, Elsevier and Wiley databases, but few using the keywords of “obesity” and “GRP78” or “BiP/HSPA5” (accessed on 15 September 2022). Most references cited are those found in Web of Science and PubMed, while some are in the Elsevier and Wiley databases between 2018 and March 2022. Early references are cited for providing the continuity and coverage of the information on the subject.

This paper focuses on the biological functions and roles of GRP78 in weight control and energy restriction via regulating mitochondrial autophagy and adenosine monophosphate-activated protein kinase-PPARγ coactivator 1α-sirtuin1 (AMPK-PGC1α-SIRT1) signal pathway (Figure 1, Pathway 1), revealing the subcellular mechanisms of GRP78 in the pathogenesis of obesity and providing new evidence for effective treatment of obesity and related diseases. 

## 2. Structure, Function, and Subcellular Localization of GRP78

GRP78, also known as immunoglobulin heavy chain binding protein BiP or human heat shock protein 5 (HSPA5), is a member of the heat shock protein 70 family (HSP70) and consists of 654 amino acids. GRP78 structurally contains a C-terminal peptide-binding domain and an N-terminal ATP-binding domain. After ATP binds GRP78, the hydrophobic peptide-binding domain is exposed and binds the misfolded protein to form a complex. Subsequently, the energy released by ATP hydrolysis is utilized to promote protein rearrangement and correct folding [10], which mediates ERS, induces apoptosis and autophagy and maintains calcium homeostasis and internal environment stability [10,11,12]. A protein having a conserved terminal amino acid sequence of EEVD, KDEL/HDEL, PEAEYEEAKK, and PECDVLDAFTDSK can bind GRP78 and reside in the cytoplasm, endoplasmic reticulum, mitochondria, and plastid, respectively. A small amount of GRP78 is expressed in the cytoplasm under normal physiological conditions, while GRP78 is transferred into the nucleus under certain pathological conditions and returned to the cytoplasm after the endoplasmic reticulum homeostasis is restored. Under certain stress, GRP78 can be actively translocated to the surface of lipid droplets [13], mitochondria, cell membrane surface, cytoplasm, and nucleus, and can also be actively secreted to the extracellular microenvironment. GRP78 even can remain in the cytoplasm by disguising its KDEL signal.

GRP78 functions in multifaceted cellular activities in different subcellular compartments. Membrane-associated GRP78 is part of receptors for virus entry and initial infection of host cells, while ER GRP78 maintains ER homeostasis and helps newly and misfolded proteins to assemble, refold and transport. Nuclear GRP78 can inhibit apoptosis induced by DNA damage. Cytoplasmic GRP78 can not only inhibit ERS and apoptotic cascade by directly inactivating caspase-7 and blocking the release of caspase-12 to the cytoplasm but also regulates viral protein assembly and suppresses tumor growth through unfolded protein responses. Recent studies found that changes in the tumor microenvironment, such as glucose deficiency, protein misfolding, hypoxia, and protein glycosylated, could promote cell-surface expression of GRP78, serving as receptors, which can be used as biomarkers and therapeutic targets for cancer and other related diseases. A better understanding of GRP78 subcellular translocation involved in obesity pathogenesis will help to reveal the subcellular mechanism of ER stress-induced obesity, which may contribute to exploiting new effective controlling measures.

## 3. GRP78 Is Increased in Patients with Obesity and Is a Prognosis Marker

GRP78 regulates insulin resistance in diet-induced obesity, which is closely associated with obesity, type 2 diabetes and cardiovascular diseases. Serum level of GRP78 and mRNA expression in human subcutaneous and omental adipose tissues of obese patients is significantly higher than those of healthy groups, and GRP78 expression is positively correlated with body mass index (BMI) [14]. Therefore, it can be used as a biomarker to predict obesity, type 2 diabetes and cardiovascular diseases [14,15,16,17]. Another clinical study involved 405 patients (obese: 52.5%; Type 2 diabetes: 68.9%; metabolic syndrome: 78.6%) attending the diabetic clinic of the Komfo Anokye Teaching Hospital. The test data showed significantly higher content of serum GRP78 in obese, type 2 diabetes and metabolic syndrome patients as compared to healthy people. Moreover, serum GRP78 is positively correlated with the value of triglyceride fatty acid (TG), non-high-density lipoprotein (HDL)-cholesterol, low-density lipoprotein (LDL)-cholesterol, carotid intima-media thickness (CIMT), and atherosclerosis index in patients [14,16]. In addition, adipose *Grp78*-knockout mice showed that GRP78 plays an essential role in adipogenesis, lipogenesis and postnatal growth in mice, and inhibition of GRP78 expression can significantly reduce lipid content in mice [18].

## 4. Roles of GRP78 in Regulating Lipid Metabolism

### 4.1. GRP78 Promotes Adipogenesis and Lipogenesis

In a high-energy diet state, adipogenesis contributes to the increased adipose tissue mass of obesity. On the contrary, adipogenesis failure makes it unable to differentiate into enough mature adipocytes which can absorb, utilize, and store excess energy, thereby inducing diabetes, etc. [19]. Inhibition of adipogenesis can effectively prevent obesity. Peroxisome proliferator-activated receptor γ (PPARγ), a nuclear transcription factor, plays an important role in adipogenesis, and its molecular targets include CCAAT/enhancer-binding protein α (C/EBPα), perilipin (PLIN), hormone-sensitive lipase (HSL) and glucose transporter type 4 (GLUT4), which participate in adipogenesis, lipid metabolism and glucose metabolism, respectively [20,21] (Figure 1 Pathway 2 & 3). As compared with wild-type, PPARγ expression in *Grp78* knockout mouse embryo fibroblasts is reduced [18]. GRP78 overexpression can promote the expression of PPARγ. Heat shock protein family (hsp70) member 12A (HSPA12A), a homolog of GRP78, is also one member of HSP70. Studies have found that HSPA12A regulates the transcription of PPARγ through a positive feedback loop, sustaining overexpression of PPARγ and maintaining the phenotype of adipocytes after differentiation (Figure 1 Pathway 4) [22].

Shown by drug affinity responsive target stability (DARTS) and surface plasmon resonance (SPR) results, epigallocatechin gallate (EGCG), dihydromyricetin (DHM) and berberine, the major active ingredient of flavonoids, induce white fat tissue browning but prevent adipogenesis or obesity via directly binding to GRP78 [23]. The dissociation constants (KD) of DHM and EGCG binding GRP78 were 22 µM and 6 µM, respectively, with significant anti-obesity activity with a maximum half effective concentration (EC_50_) of 400 µM and 75 µM, respectively [24].

GRP78 plays an important role in adipogenesis, lipogenesis, metabolic homeostasis, fetal and postnatal growth and development of mice [18]. Knockout *Grp78* in mouse embryonic fibroblasts, 3T3-L1 cells and adipocyte tissue showed adipogenesis and lipogenesis problems. The aP2-cre-mediated GRP78 deletion leads to a reduction of lipoatrophy in gonadal and subcutaneous white adipose tissue and brown adipose tissue by up to ∼90%, severe growth retardation, bony defects and grossly expanded endoplasmic reticulum in white adipose tissue. However, plasma triglyceride levels and plasma glucose and insulin levels are reduced by 40–60% as compared to wild-type mice, suggesting an improvement of the insulin sensitivity in *Grp78* knockout mice. The results indicated that GRP78 is essential for adipogenesis in vivo. Unexpectedly, the mutant mice showed early postnatal death and unique distinction from previously characterized lipodystrophic mouse models.

Sterol regulatory element binding protein-1c (SREBP-1c) is a transcription factor that critically regulates lipid metabolism. Insulin-induced cleavage and activation of SREBP-1c, which is the cause of ectopic fat deposition in the liver. SREBP-1c directly binds GRP78, which remains in the endoplasmic reticulum without transcription activity. Nevertheless, dissociation of SREBP-1c and GRP78 promotes the transport of the SREBPs-SCAP complex to Golgi, where SREBP-1c is cleaved, and then active SREBP-1c is transferred into the nucleus to initiate expression of genes involved in triglyceride and cholesterol synthesis (Figure 1 Pathway 5). Therefore, triglyceride and cholesterol levels are significantly increased [25,26,27]. Through hepatic overexpression of GRP78 in *ob*/*ob* mice using an adenovirus vector, it was found that GRP78 overexpression inhibits SREBP-1c cleavage and the expression of SREBP-1c target genes lowers liver triglycerides and cholesterol levels and improves insulin sensitivity [25].

Intermittent fasting can induce pregnancy zone protein (PZP) production and release in the liver, followed by the translocation of GRP78 to the cell surface of brown adipose tissue (BAT). The binding between PZP and GRP78 upregulates UCP1 expression through the p38 mitogen-activated protein kinases-activating the transcription factor-2 (p38 MAPK-ATF2) signaling pathway and promoting the thermogenic metabolism of BAT. These results indicated that GRP78 is an indispensable regulator of PZP-induced thermogenesis [28]. (Figure 1 Pathway 6).

### 4.2. GRP78 Promotes De Novo Formation of Lipid Droplets

Lipid droplets are the main cellular sites for triglycerides and other lipids storage [29]. Lipid droplet fusion and growth are closely regulated by lipid droplet-coated proteins adapted for cellular energy needs. Proteomic studies have found that HSP70 proteins, including GRP78, are major structural proteins of lipid droplets [30]. Expressed GRP78 is translocated to lipid droplets in rat adipocytes upon heat stimulation. This process occurs neither in a temperature-dependent nor time-dependent manner but occurs suddenly in 30–40 min and rapidly reaches a stable state within 1 h at 40 °C heat shock.

Although GRP78 is co-localized with phospholipids on the surface of lipid droplets, co-immunoprecipitation experiments did not show direct interactions between GRP78 and phospholipids. Alkaline treatment indicated an association of GRP78 with the surface of the droplets through non-hydrophobic interactions. Therefore, it is speculated that GRP78 may non-covalently associate with monolayer microdomains of lipid droplets in a manner like its interaction with lipid bilayer moieties composed of specific fatty acids. As an acute cell-specific response to heat stimulation, the accumulation of GRP78 on adipocytes lipid droplets may be involved in the stabilizing of droplet monolayer phospholipid, transferring or chaperoning denatured proteins to the lipid droplets for subsequent refolding.

Reticular protein 3 (RTN3) plays a key role in regulating triglyceride synthesis, storage, and lipid droplet fusion. Studies have found that RTN3 enhances SREBP-1C and AMPK activity through its interactions with GRP78 and leading to obesity and hyperlipidemia [31] (Figure 1 Pathway 7). SREBP-1c and AMPK are downstream of GRP78. Berbamine inhibits GRP78 and also induces AMPK activation, which regulates the mammalian target of the rapamycin/ SREBP-1c (mTOR/SREBP-1c) axis and the nuclear factor E2-related factor 2 (Nrf2)/antioxidant response element (Nrf2/ARE) pathway to allay lipid accumulation and oxidative stress in steatotic HepG2 cells. [32]

### 4.3. GRP78 Negatively Regulates Mitochondrial Biosynthesis and Energy Balance

A decrease in mitochondrial numbers and dysfunction can lead to obesity. GRP78 regulates mitophagy, mitochondrial biogenesis and energy balance [33,34]. Mitophagy is inhibited by GRP78 down expression, which is a vital way to activate browning and prevent obesity [35,36,37]. Further research found potential mitophagy mediation by GRP78 through the AMPK/mTOR signaling pathway, which leads to that energy intake exceeding expenditure and obesity. It was also found that GRP78 overexpression in the mitochondria triggers PINK1/IP3R-mediated neuroprotective mitophagy [38].

### 4.4. GRP78 Causes Insulin Resistance

Insulin resistance is considered an underlying etiology of metabolic syndrome and cardiovascular disease associated with obesity, such as type 2 diabetes. Both human and animal experiments have shown a positive correlation between GRP78 levels and insulin resistance [14,39] and effective improvement in insulin sensitivity and glycemic control via GRP78 down-regulation [15].

Further studies showed that GRP78 could activate the insulin signaling pathway and improve insulin sensitivity through phosphorylation modification of protein kinase B (Akt) [40]. GRP78 down-regulates Akt expression and phosphorylation but does not directly affect upstream pyruvate dehydrogenase kinase 1 (PDK1) activity. PDK1 is a protein kinase B (PKB), also known as Akt, which is a serine/threonine-specific protein kinase. Co-immunoprecipitation of GRP78 and p-Akt (Ser473) immune-complex consists of non-phosphorylated Akt (Ser473 and Thr308) (Figure 1 Pathway 8). In-situ proximity ligation analysis showed co-location of GRP78 with Akt in cell membrane after ER stress induction and an increase in phosphorylation of Akt Ser473 but a decrease (i.e., inhibition) of Thr308. siRNA-mediated GRP78 knockdown enhances phosphorylation at Ser473 by 3.6-fold, but no impact at Thr308 [40].

Human jejunal mucosa secretes GRP78 in vitro, and bariatric surgery improves insulin resistance and type 2 diabetes by reducing intestinal GRP78 secretion. Plasma GRP78 levels in insulin resistance patients are higher than in healthy people and those who returned to normal physiology after duodenal jejunal bypass surgery, and plasma GRP78 level was negatively correlated with insulin sensitivity but positively correlated with body mass index (BMI) [41].

A high-calorie diet can increase plasma GRP78 levels and induce insulin resistance. GRP78 stimulates the accumulation of lipid droplets and inhibits Akt Ser473 phosphorylation and glucose uptake in both immortal liver cells and peripheral blood plasma cells. However, a converse phenomenon occurs when GRP78 serum levels decrease or insulin-resistant patients undergo duodenal jejunal bypass surgery. Intestinal secretion of GRP78 may be the cause of insulin resistance, and duodenal jejunal bypass surgery may reduce GRP78 secretion and improve insulin sensitivity by shortening food transportation or reducing lipid stimulus released from endocrine cells [41]. GRP78 is essential for proinsulin synthesis, and up-regulation of GRP78 can increase insulin secretion in response to hyperglycemia, while down-regulation of GRP78 can decrease insulin secretion and lead to significantly low levels of insulin [42].

### 4.5. GRP78 Can Eliminate Liver Lipotoxicity and then Improve Liver Steatosis

ER stress plays an important role in hepatic steatosis and insulin resistance in obese mice models [43]. GPR78 is involved in the pathogenesis of nonalcoholic fatty liver disease and is associated with hepatic steatosis, insulin resistance, inflammation, and apoptosis [44,45]. GRP78 plays a key role in maintaining lipid balance in the liver, and GRP78 overexpression can reduce the hydrolysis of SREBP-1c induced by ER stress and liver steatosis [43]. It was also reported that GRP78 overexpression in HepG2 cells prevents ER stress and cytotoxicity induced by palmitic acid, and further studies found that GRP78 may reduce lipid peroxidation and damage induced by oxidative stress [46]. Some studies suggest that GRP78 overexpression in the liver can reduce the expression of SREBP-1c, reduce ectopic triglyceride deposition in the liver, and enhance insulin sensitivity of *ob*/*ob* mice [25].

## 5. Molecular Mechanism of Action of GRP78 during Obesity Development

### 5.1. Proteins and Small Molecules Directly Bind GRP78

New data on the therapeutic target GRP78 for obesity and its effectiveness are still rare. Ingenuity Pathway Analysis (IPA) analysis showed the relevance of 1216 proteins to obesity and the direct interactions of 139 of them with GRP78 (Figure 2). Studies such as co-immunoprecipitation (co-IP), fluorescence resonance energy transfer assay (FRET), GST pull-down technique, microscale thermophoresis (MST), and bioinformatics analysis have indicated direct interactions between GRP78 and downstream target proteins such as PPARγ, SREBP-1 and p-Akt [47] (Table 1). Research shows that binding of α2-macroglobulin-GRP78 triggers the expression of IP_3_, RAS and MAPK, PAk2, PI3K, PLD, COX2, and cPLA2 and also increases the intracellular second messengers (Ca^2+^ and cAMP) level and pH value, which can activate the downstream signal of PPARγ/C/EBPα, PLIN/HSL, etc. (Figure 1 and Figure 2). A secreted form and the cell surface localization of GRP78 bound to phosphoinositide 3-kinases (PI3K) then induced Akt activity to activate the PI3K/Akt/mTOR signaling pathway. GRP78 and Akt form a positive feedback loop [48], and continuous overexpression of mTOR can further enhance triglyceride synthesis and obesity (Figure 1) [49].

Nucleotide-binding domain (NBD) regulates the translocalization and secretion of GRP78. GRP78 translocation mechanism is related to its conserved terminal amino acid sequence, which is discussed in Section 2. Numerous studies using surface plasmon resonance (SPR) and GRP78 ATPase analysis have verified that EGCG and other compounds bind the nucleotide-binding domain of GRP78 (Figure 3). Five active ingredients (salvianolic acid A, naringin, platycodon, platycodon D, diosmin and isovitexin) were identified and extracted from 51 traditional Chinese medicines, which may directly interact with GRP78 [47,50]. EGCG, DHM, berberine and structural analogs have been discovered through drug screening for their inhibition against tumor-secreted GRP78 (Table 2 and Table 3). It is known that direct interactions between GRP78 and the 12 molecules are anti-Glioblastoma (GBM) (Table 2).

**Figure 2 ijerph-19-15965-f002:**
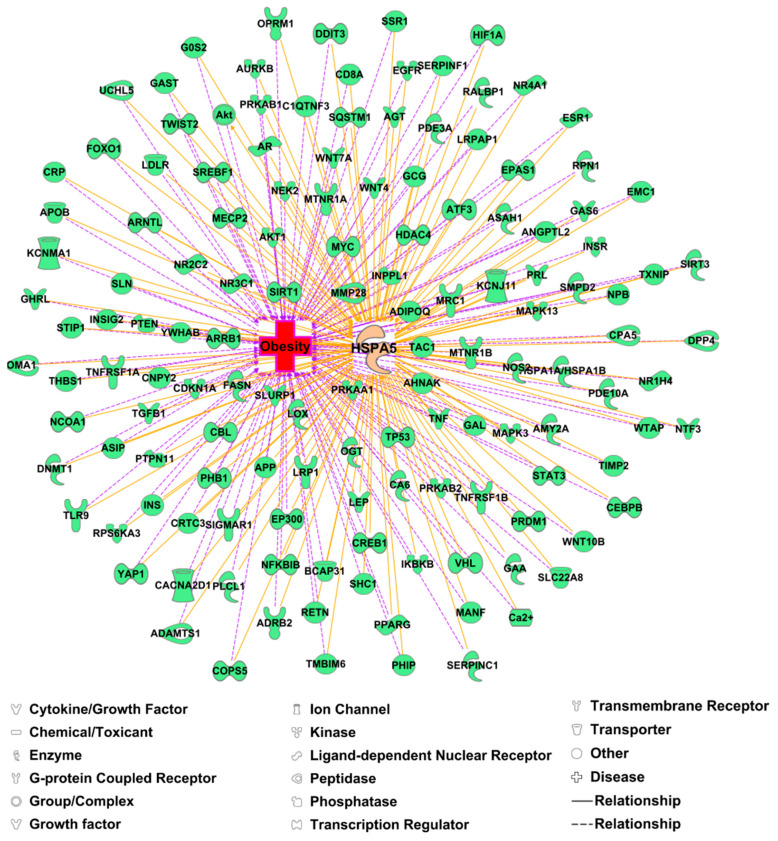
Ingenuity Pathway Analysis (IPA) result of GRP78 (HSPA5) with its interacting proteins relevant to obesity. Using “obesity” as the keyword, 1216 proteins, including HSPA5 (GRP78), were obtained from the IPA database. According to the analysis of the “Build-Path Explorer” module, 139 molecules have interactions with HSPA5.

**Table 1 ijerph-19-15965-t001:** List of proteins that directly bind GRP78 [47,51,52,53,54,55,56,57,58,59,60,61,62,63,64,65,66,67,68].

No.	Protein	Binding Site and Interaction Mode	Downstream Signaling/Effect	Methods
1	PERK/IRE1/ATF6	Calnexin and vimentin interact indirectly with annexin A2 via GRP78	Initially activate PERK-eIF2A, IRE1-XBP1 and ATF6 signal pathways and trigger cell apoptosis	IP-MS; Bioinformatics analysis
2	PERK	Disinhibited by release from GRP78, and then phosphorylates eIF2α	Activated phosphorylation of eIF2α by PERK inhibits translation initiation, decreases protein synthesis and protein influx in the ER, then phosphorylation of ATF4 activates CHOP and apoptosis by PERK-eIF2α-ATF4-CHOP pathway	IHC
3	IRE1	Disinhibited by release from GRP78	Splicing mRNA encoding XBP1 triggers endoribonuclease activity of IRE1, and then targets genes in protein folding and ERAD by IRE1-TRAF2-JNK pathway	IHC
4	ATF6	GRP78 directly activates ATF6 by binding to its luminal domain and inhibiting its Golgi localization signals; ATF6 is disinhibited by release from GRP78, then was cleaved and activated at Golgi apparatus	Active ATF6 moves to nucleus and upregulates proteins that promotes ER protein folding; ATF6 downstream target gene GRP78	IFA; IP.
5	SREBP-1	GRP78 retains SCAP/SREBP1 in the ER via direct interaction	Inhibit PI3K/Akt pathway; Positively regulates the transcription of ACC and FASN; SREBP-1 downstream targets involved in fatty acid synthesis, including FASN	Co-IP
6	CNPY2	CNPY2 combines with GRP78 under normal conditions; After UPR inducer chlamycin treatment, CNPY2 disassociates from GRP78 and binds to PERK	Activate PERK-CHOP signaling; CNPY2 blocked the PERK-CHOP pathway of the unfolded protein response	*Cnpy2* knockout mice fed a high-fat diet
7	p-Akt	Exocrine GRP78	Activate PI3K/AKT signaling, then promote tumor cell proliferation and decrease the sensitivity of HCC cells to sorafenib	Co-IP
8	SR-A	SR-A binds to cytoplasmic GRP78	GRP78 involved in SR-A mediates lipid endocytosis; Inhibits inflammatory cytokine expression (TNF-α, IL-1) through MAPK, PI3K-Akt and NF-κB signaling	IP; Indirect IFA; FRET
9	SCNN1B	SCNN1B interacts with GRP78 and induces GRP78 degradation via polyubiquitination.	Increase ubiquitin-mediated degradation of GRP78, subsequently trigger the unfolded protein response (UPR)	Tissue microarray analysis; SCNN1B ectopic expression and knockdown; IFA; IP–MS.
10	Anti-EGFR antibody	Anti-EGFR antibody combines and co-locates with GRP78	Block the promotion of GRP78 to the invasion of cancer stem cells	Transwell; Confocal microscopy; WB

IP: Immunoprecipitation; MS: mass spectrometry; SR-A: Class A scavenger receptor; IHC: Immunohistochemistry; IFA: Immunofluorescence; FRET: Fluorescence resonance energy transfer; WB: Western blot.

**Figure 3 ijerph-19-15965-f003:**
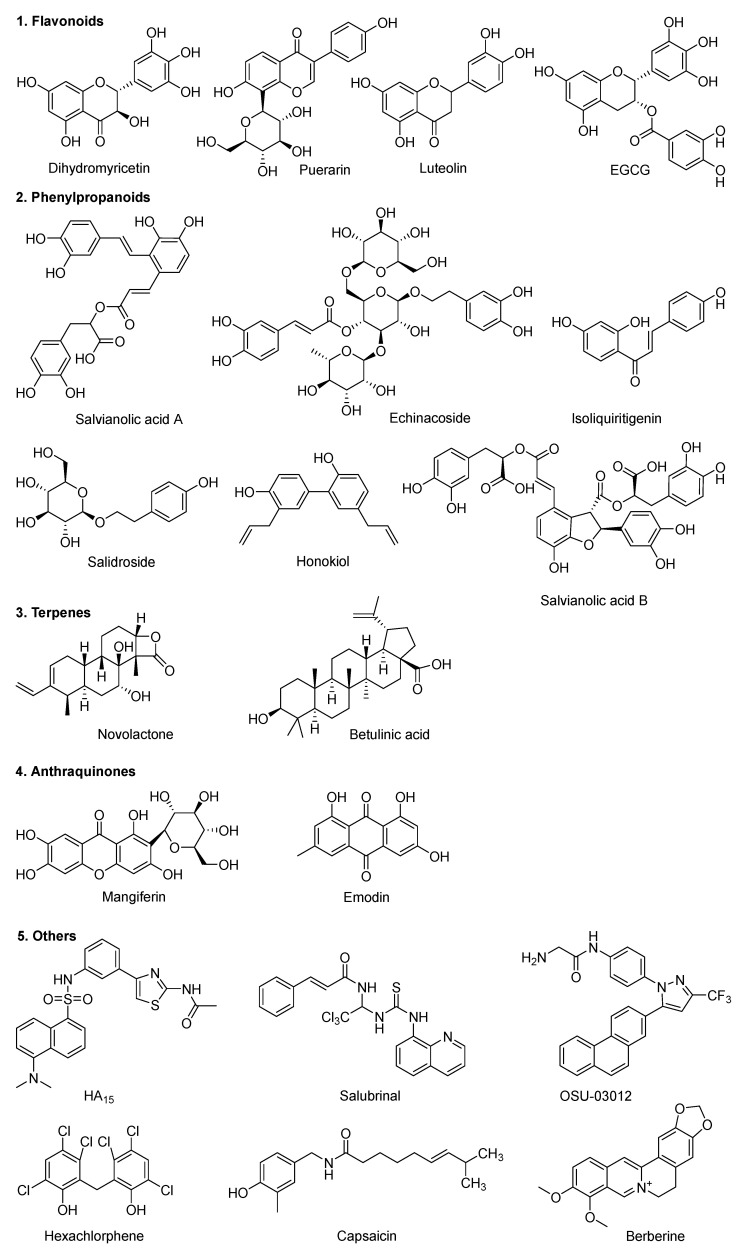
GRP78 inhibitors targeting its ATPase/nucleotide-binding domain.

**Table 2 ijerph-19-15965-t002:** GRP78 inhibitors and inducers and their action mechanisms and effects [51,52,53,54,55,56,57,58,59,60,61,62,63,64,65,66,67,68].

No.	Compound/Combination Therapy in GBM	Action Mechanism	Effects	Model
1	EGCG; EGCG+TMZ/5-fluorouracil/taxol/vinblastine/gemcitabine/TRAIL/doxorubicin/paclitaxel/IFN-α2b	GRP78 (NBD)	Impair GRP78 function; Enhance cytotoxicity when used with TMZ or others	Human cell lines, in vivo [51,52]
2	Honokiol; Honokiol+TMZ/fenretinide/bortezomib	GRP78 (NBD)	Interfere with GRP78 folding; Induce ER stress-mediated apoptosis with TMZ	Human cell lines [53,54,55]
5	NEO100 (Clinical trials); NEO100+TMZ/DMC/relfinavir	ER stress	Disrupt survival pathways; Induce more apoptosis with TMZ and others, reduce GBM invasion capacity, prolong survival	Human cell lines, in vivo [58,59,60]
8	EGF-SubA; EGF-SubA+radiation+TMZ	Cleave GRP78	Correct the ATPase and protein binding domains; Delay tumor growth, enhance effects of TMZ and ionizing radiation	Human cell lines in vivo mouse models [64]
3	OSU-03012; radiotherapy+OSU-03012;	GRP78 (NBD)	PDK1 inhibition, GRP78 inhibition, PERK signaling inhibition; Enhance radiosensitivity; Prolong survival	Human cell lines, in vivo mouse models [52,56]
4	Celecoxib and bortezomib Celecoxib+bortezomib+GRP78 inhibition	ER stress	Augment ER stress; Induce ER stress-mediated apoptosis	Human cell lines [57]
6	HA15	Bind and inhibit GRP78	Disrupt GRP78 complexes with PERK/IRE1/ATF6; Induce apoptosis	Human cell lines, in vivo mouse models [61,62]
7	IT-139	GRP78	Involve transcriptional and post-transcriptional mechanisms; Decrease therapeutic resistance	Human cell lines, in vivo human xenograft studies [63]
8	EGF-SubA TMZ+radiation therapy+EGF-SubA [64]	GRP78; Cleave GRP78	TMZ and ionizing radiation; Delay tumor growth, enhanc effects of TMZ and ionizing radiation	Human cell lines in vivo mouse models
9	Anti-GRP78 antibody; Ionizing radiation+anti-GRP78 antibody	Bind to surface GRP78	Enhance effects of ionizing radiation via suppression of PI3K/AKT/mTOR signaling, and result in tumor delay	Human cell lines, in vivo mouse xenograft models [65]
10	RGD ligand-directed phage with GRP78 promoter	Bind GRP78	RGD tumor homing ligand binds and improves expression of therapeutic transgenes with GRP78 promoter	Human cell lines in vivo [66]
11	TMZ-induced AAV phage with GRP78 promoter; TMZ+phage	RGD4C/AAV/phage/GRP78 binding	Activate therapeutic transgenes expression with GRP78 promoter; Permit dose escalation of TMZ	Human cell lines, mouse xenograft models [67]
12	GIRLPG; Radiation+phage	Bind GRP78	Allow for adenovirus-mediated gene delivery to target tumor cells; Enhance radiation therapy and therapeutic transgenes expression	Human cell lines, mouse xenograft models [68]

GBM: Glioblastoma; NBD: Nucleotide-binding domain; TMZ: Temozolomide, which is the leading compound in GBM treatment; TRAIL: Tumor necrosis factor (TNF)-related apoptosis-inducing ligand; IFN-α2b: interferon-α2b; NEO100: Perillyl alcohol (monoterpene); EGF-SubA: Epidermal growth factor; OSU-03012: Celecoxib derivative; GIRLPG: GRP78 binding peptide.

**Table 3 ijerph-19-15965-t003:** GRP78_ATPase_ nucleotide affinities (K_D_) determined by surface plasmon resonance (SPR) [6,23,47,50,52,53,54,55,56,69,70,71,72,73,74,75,76,77,78].

No.	Chemicals	2 mM MgCl_2_	K_D_ (M) ^a^	5 mM EDTA	Biologic Activity/Reference
			Untreated ^b^		
1	ATP	(4.5 ± 2.9) × 10^−7^	(7.8 ± 7.1) × 10^−7^	(9.8 ± 4.4) × 10^−6^	[69]
2	ADP	(1.2 ± 0.9) × 10^−8^	(2.7 ± 4.4) × 10^−7^	(4.3 ± 7.7) × 10^−5^	[69]
3	7-deazaATP	(3.0 ± 2.0) × 10^−8^	(1.5 ± 0.9) × 10^−7^	(9.0 ± 5.4) × 10^−7^	[69]
4	AMPPCP	(5.9 ± 1.2) × 10^−5^	(5.2 ± 4.1) × 10^−5^	>1 × 10^−3^	[69]
5	2′-deoxyATP		(7.5 ± 5.0) × 10^−4^	>1 × 10^−3^	[69]
6	Honokiol				Bind GRP78 (NBD) using DSC and ITC [53,54,55,70]
7	Mangiferin				[71]
8	Isoliquiritigenin				[72]
9	OSU-03012				[56,72,73]
12	Luteolin				[74]
13	DHM		22 × 10^−6^		Anti-adipogenesis, EC_50_ 284 μM [24]
14	HA15				Induce UPR and kill BRAF mutant melanomas [75]
15	EGCG		6 × 10^−6^		bind GRP78 (NBD); Anti-adipogenesis, EC_50_ 103 μM [52,72,73,76]
16	Salvianolic acid A				Lysine 633 acetylation of GRP78 to block GRP78 secretion [72,77]
17	Salvianolic acid B				[74]
18	Salidroside				[74]
19	Salubrinal				[74]
20	Echinacoside				[74]
21	Betulinic acid				[72]
22	Capsaicin				[6]
23	Berberine				[6,23,47,50]
24	Naringin				Inhibit the expression of GRP78 [77]
25	Platycodon platycodon D				Up-regulate expression of GRP78 [77]
26	Diosmin				[77]
27	Isovitexin				[77]
28	Emodin				[6]
29	Curcumin				DARTs, directly targeting GRP78 [6]
30	Novolactone				Destabilize HER2 and EGFR in cancer cells [75]
31	Rifampicin				[74]
32	Puerarin				[6]
33	Hexachlorphene				Induce apoptosis and block autophagy in melanoma cell lines [78]
34	VER-155008		80 × 10^−9^		[72]

^a^ K_D_ values are presented as mean ± SD of three independent experiments (*n* = 3); ^b^ Untreated denotes that SPR running buffer contains no addition of either MgCl_2_ or EDTA. DSC: Differential scanning calorimetry; ITC: Isothermal titration calorimetry analysis; DHM: Dihydromyricetin; EGCG: Epigallocatechin gallate.

### 5.2. GRP78 Monomer/Heteromer and Conformational Changes

Over the past few decades, research attention has been paid to the targeted therapy, transcriptional activity, unfolded proteins response, ATP binding and hydrolysis activities of GRP78 (Figure 4) [72]. Subsequently, a great deal of scientific research has been devoted to clarifying the functions of GRP78 in both in vitro and in vivo. GRP78 was previously thought to exist as a monomer in eukaryotic cells and spontaneously dimerize/oligomerize to function. However, recent studies showed that GRP78 exists as a dynamic pool of monomer, dimer and oligomer, and dimer/oligomer transition from active monomer resulting in inactive GRP78 form [10]. EGCG, DHM and HA15 exert anti-obesity activity by inducing GRP78 conformational change [50]. Since ER stress-regulated by GRP78 is essential for virtually all cellular activities, either intracellular signaling or transcriptional regulation, and its association with several pathologies. Therapeutics targeting GRP78 via reducing its expression might have side effects. The potential of GRP78-targeted anti-tumor drugs has no effect on intracellular GRP78 expression but blocks its secretion. GRP78 conformation transition only works under specific pathological conditions, and allosteric modulation of GRP78 has better specificity. However, the mediation of GRP78 conformational transitions remains a gap, and the mechanism still is not clear.

Allosteric modulators are appealing sources of drug discovery for sensitivity, selectivity and security, which can be used in an ultra-low dose and do not fully close substrate channeling. Nevertheless, traditional regulators need higher doses to compete with substrates for the active site. However, GRP78 allosteric modulators still face challenges, such as no allosteric prediction models, lack of knowledge of the quaternary structure-activity relationship, low affinity, unknown certain binding sites, intracellular/extracellular distribution regulatory, incorporation polar and Log *P* median solubilization to enhance bioactivity.

Nevertheless, the great advantages of high selectivity and safety have raised a growing interest in the allosteric regulatory molecules of GRP78 in recent years. Although binding sites and crystal structures of allosteric regulatory molecules of GRP78 have been reported, there is a need for drug development and structural biology research. In addition to traditional experimental methods, computational biology and bioinformatics methods are powerful tools for the development of allosteric modulators targeting GRP78.

It is of great significance to explore the relationship between biological phase separation and disorder proteins in human diseases. Protein phase separation disorder can lead to protein transition from reversible liquid-liquid phase to irreversible liquid-solid phase and form irreversible amyloid-like aggregations.

It was first found that the chaperone GRP78 regulates liquid-liquid phase separation (LLPS) and liquid-solid phase transition through its own phase separation, which can maintain the LLPS homeostasis of transactive response DNA binding protein 43 (TDP-43) and inhibit spontaneous TDP-43 aggregation into amyloid fibrils in neurons under stress. Lipid droplet biogenesis is also driven by LLPS [79,80]. Phase separation is a basic organization in the cell membrane, and allosteric regulators may play a role by interfering with chaperones’ phase separation [79,80]. Although GRP78 is constitutively expressed, its synthesis can be induced by a variety of stresses, such as neurotoxicity, myocardial infarction, and arteriosclerosis. Its role is double-faceted, which cannot be clearly explained from the genome and transcription level. Protein post-translational modifications of GRP78 include phosphorylation, glycosylation, ubiquitination, allosteric modulation and acetylation [73], which can dynamically control the downstream proteins undergoing phase separation in multi-level. It warrants an exciting direction for future research.

## 6. Conclusions

The roles of GRP78 on obesity are clear, including promoting adipogenesis and lipogenesis, promoting de novo formation of lipid droplets, negatively regulating mitochondrial biosynthesis and energy balance, causing insulin resistance, eliminating liver lipotoxicity and then improving liver steatosis. While GRP78 shows regulatory functions against obesity, drug engagement needs further research. The majority of these natural ingredient studies have not yet been tested in clinical trials. Current in vitro and in vivo animal studies show promising anti-obesity effects. Molecules like ingredients of herbs described in this review may provide clues to future studies. Allosteric modulators with high efficacy at an ultralow dose and LLPS functions of GRP78 are probably ways to alleviate obesity.

## Figures and Tables

**Figure 1 ijerph-19-15965-f001:**
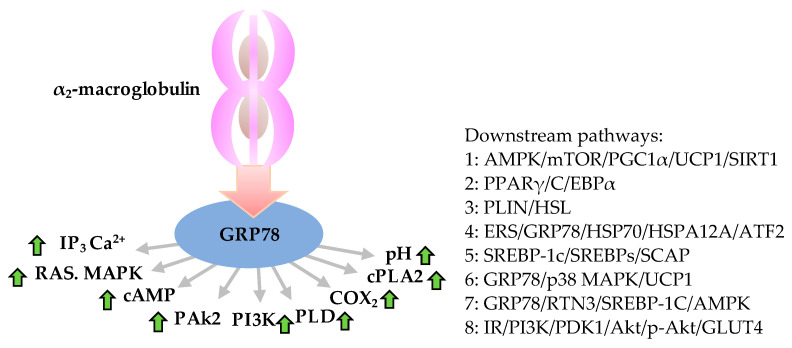
Anti-obesity role of GRP78 and downstream molecular functions associated with obesity. Roles and downstream pathways of GRP78 relevant to obesity initiated by binding of α2-macroglobulin-GRP78 that triggers the expression of IP_3_, RAS and MAPK, PAk2, PI3K, PLD, COX2, and cPLA2 and increases concentrations of cAMP, Ca^2+^ and pH value (**left**), which can activate the downstream pathways (**right**). Green arrows indicate up-expression. AMPK, adenosine monophosphate (AMP) activated protein kinase; Akt, an ubiquitous serine/threonine kinase, also known as protein kinase B (PKB) or RAS-alpha; ATF2, activating transcription factor-2; cAMP, cyclic adenosine monophosphate; C/EBPα, CCAAT/enhancer-binding protein α (which CCAAT is a distinct pattern of nucleotides with GGCCAATCT consensus sequence that occurs upstream by 60–100 bases to the initial transcription site); COX_2_, cicloxigenases; cPLA2, cytosolic phospholipase A 2; ERS, endoplasmic reticulum stress; GLUT4, glucose transporter type 4; GRP78, 78 kDa glucose-regulated protein; HSP70, heat shock protein 70 family; HSL, hormone sensitive lipase; HSPA12A, heat shock protein family a (hsp70) member 12A; IP3, inositol triphosphate; IR, insulin resistance; mTOR, mammalian target of rapamycin; PAk2, p21-activated protein kinase; PGC1α, PPARγ coactivator 1α; PDK1, pyruvate dehydrogenase kinase 1; PLD, phospholipase D; PI3K, phosphoinositide 3-kinases; PPARγ, peroxisome proliferator activated receptor γ; PLIN, perilipin; p38 MAPK, p38 mitogen-activated protein kinases; p-Akt, phosphorylated Akt; RAS, renin-angiotensin system; RTN3, reticular protein 3; SIRT1, sirtuin1; SREBP-1c, sterol regulatory element binding protein-1c; SREBPs, sterol regulatory element binding protein; SCAP, SREBP cleavage-activating protein; TG, triglyceride fatty acid; UCP1, uncoupling protein 1.

**Figure 4 ijerph-19-15965-f004:**
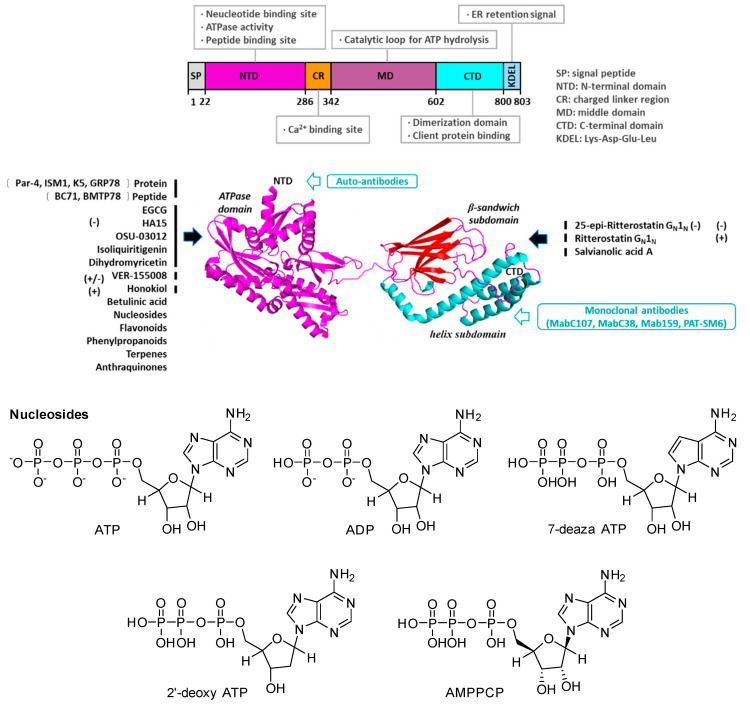
Regulatory functions of GRP78 and downstream molecular functions relevant to anti-obesity. Domain structures (**top**) and 3D structural model (**middle**) of GRP78, molecules interacting with either the ATPase domain (in magenta), β sandwich subdomain (in red) or the helix subdomain (in cyan) of GRP78, and chemical structures of nucleosides (**bottom**). The cited small molecules can up (+) or down (−) regulate GRP78. Modified from [72].

## Data Availability

The data supporting the conclusion of this article will be available upon request from the corresponding authors.

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
