# Peer review of "GRP78 Activity Moderation as a Therapeutic Treatment against Obesity"

_ijerph, 2022, doi:10.3390/ijerph192315965_

Round 1
Reviewer 1 Report
This review article summarized the current understanding of the role of GRP78 on metabolic health and obesity. The authors have discussed the mediation of GRP78 in a number of obesity-related signaling pathways that regulate lipid metabolism and metabolic homeostasis and how the disruption of these signaling pathways due to the change in GRP78 level or activity leads to the development of obesity and metabolic abnormality. Overall, this review is very comprehensive and well-written. I have a few comments that may improve the manuscript.
The font style is not consistent in the manuscript.
This manuscript focused on discussing GRP78 as a treatment target for obesity and metabolic abnormalities. The authors mentioned the two approved drugs for treating obesity stated in line 27. The authors also mentioned a list of hypotheses of obesity development in line 36-39. Which hypothesis does the currently approved drug target? Notably, the authors mentioned a list of Chinese medicines commonly used to alleviate obesity. What is the purpose of mentioning these Chinese medicines? Are their effects, at least partly, mediated through GRP78 (or possibly mediated through GRP78) or through a different pathway? Do the authors mean the currently available approved medicines are not sufficient and therefore urge the discovery of new treatment targets for obesity and the Chinese medicine may provide some clues (as the authors mention some flavonoids and anthraquinone interact with GRP78)? While addressing these comments, the authors may need to rearrange the first two paragraphs in the introduction section to make the flow smooth.
In line 39, it is not clear what “with this hypothesis” refers to.
The authors may want to further discuss why the ERS hypothesis is important among other hypotheses. How does it relate to obesity development and link obesity to other metabolic abnormalities? The authors may need to further explain the ERS hypothesis.
Does GRP78 mainly target the ERS hypothesis? Any other hypothesis involved?
This review has comprehensively summarized how GRP78 affects metabolic health. It has been demonstrated that the serum level of GRP78 was associated with obesity, insulin resistance, and metabolic syndrome. GRP78 sounds like a potential treatment target. However, it seems that the effects of GRP78 are very sophisticated. In addition to being the master regulator of ER stress, GRP78 has different functions in different locations of the cell or body. Moreover, GRP78 appears to be the upstream mediator of a number of important signaling pathways. A simple general downregulation of GRP78 as a whole may lead to harmful effects on the body while upregulation in certain locations may bring benefits. Section 4.6 have mentioned some other possible options, such as allosteric modulation, to target GRP78 for treating obesity and metabolic abnormalities. Notably, the authors specifically mentioned the example of Chinese medicine for alleviating obesity in the introduction section and pointed out that there are five active ingredients extracted from 51 Chinese medicine that may interact with GRP78 in section 4.6. However, they did not further discuss the use of medicine in regulating GRP78. While there are ingredients that interact with GRP78, are these Chinese medicines used in treating obesity or metabolic disorders? Other than being a sub-section in section 4. I suggested making this sub-section a new section discussing the difficulty of using GRP78 as a treatment target, possible modulation of GRP78 on obesity and metabolic abnormality treatments (eg conformational change and targeting specific regions or organs), and the future direction of research and drug development.
Author Response
- The font style is not consistent in the manuscript
R: Changes have been made throughout the entire manuscript as instructed.
- This manuscript focused on discussing GRP78 as a treatment target for obesity and metabolic abnormalities. The authors mentioned the two approved drugs for treating obesity stated in line 27. The authors also mentioned a list of hypotheses of obesity development in line 36-39. Which hypothesis does the currently approved drug target? Notably, the authors mentioned a list of Chinese medicines commonly used to alleviate obesity. What is the purpose of mentioning these Chinese medicines? Are their effects, at least partly, mediated through GRP78 (or possibly mediated through GRP78) or through a different pathway? Do the authors mean the currently available approved medicines are not sufficient and therefore urge the discovery of new treatment targets for obesity and the Chinese medicine may provide some clues (as the authors mention some flavonoids and anthraquinone interact with GRP78)? While addressing these comments, the authors may need to rearrange the first two paragraphs in the introduction section to make the flow smooth.
R:
(1) The action mechanism of two approved drugs for treating obesity stated in line 27 have been added.
(2) A list of hypotheses of obesity development in line 36-39 not just for clinical drugs, but also for other molecules in clinical and preclinical trials. The action mechanism and targets of those molecules are unclear, which may be a reason for their failure to enter clinical application. Some of them mediated through GRP78 (like tangeretin; resveratrol; quercetin; EGCG), or through different pathway, such as PPARγ/C/EBPα, SREBP/AMPK, SREBP-1c/SREBPs/SCAP; or multi-target and multi-pathway, for example, EGCG has multi-targets on GRP78, MAPK, fatty acid synthase; Honokiol has multi-targets on PPARγ, dipeptidyl peptidase IV; Dihydromyricetin has multi-targets on GRP78 and SIRT.
(3) The currently available approved medicines are not sufficient, and it is therefore necessary to discover new treatment targets for obesity and the Chinese medicine may be a good source for drug leads. The first two paragraphs in the introduction section were rearranged as suggested.
- Line 39, it is not clear what “with this hypothesis” refers to. The authors may want to further discuss why the ERS hypothesis is important among other hypotheses. How does it relate to obesity development and link obesity to other metabolic abnormalities? The authors may need to further explain the ERS hypothesis.
R: Changes have been made as suggested. There are some connections and overlaps between different hypotheses.
- Does GRP78 mainly target the ERS hypothesis? Any other hypothesis involved?
R: Yes. GRP78 is a major ER chaperone as well as a master regulator of the UPR. Energy balance hypothesis is also involved, and was described in section 4.3.
- This review has comprehensively summarized how GRP78 affects metabolic health. It has been demonstrated that the serum level of GRP78 was associated with obesity, insulin resistance, and metabolic syndrome. GRP78 sounds like a potential treatment target. However, it seems that the effects of GRP78 are very sophisticated. In addition to being the master regulator of ER stress, GRP78 has different functions in different locations of the cell or body. Moreover, GRP78 appears to be the upstream mediator of a number of important signaling pathways. A simple general downregulation of GRP78 as a whole may lead to harmful effects on the body while upregulation in certain locations may bring benefits. Section 4.6 have mentioned some other possible options, such as allosteric modulation, to target GRP78 for treating obesity and metabolic abnormalities. Notably, the authors specifically mentioned the example of Chinese medicine for alleviating obesity in the introduction section and pointed out that there are five active ingredients extracted from 51 Chinese medicine that may interact with GRP78 in section 4.6. However, they did not further discuss the use of medicine in regulating GRP78. While there are ingredients that interact with GRP78, are these Chinese medicines used in treating obesity or metabolic disorders? Other than being a sub-section in section 4. I suggested making this sub-section a new section discussing the difficulty of using GRP78 as a treatment target, possible modulation of GRP78 on obesity and metabolic abnormality treatments (eg conformational change and targeting specific regions or organs), and the future direction of research and drug development.
R: Changes have been made as recommended. 4.6 is changed to section 5.
Reviewer 2 Report
1) This paper summarizes the GPR78 inhibitors targeting its ATPase domain, small molecules and proteins that directly bind GPR78, and the putative mechanisms of GPR78 in regulating lipid metabolism. First, however, this review must highlight the importance and impact of the reference for drug development scientists.
2) The word size in the different paragraphs is not standardized.
3) Sections 2 to 4.5 explain the function, molecular mechanism, physiological role, role related to disease, pathological function…etc. Therefore, it is suitable for this review to initiate and emphasize the role of GPR78 in obesity and the value used for the drug development target. However, these kinds of reviews are too many. The authors can find many publications only delineate the physiological and pathological functions of GPR78. So, the value of this review explores the role of GPR78 as a critical modulation factor for the treatment of obesity and is the most crucial paragraph in section 4.6.
4) However, section 4.6 delineates the protein and small molecules that directly bind with GPR78 and the molecular mechanism of GPR78 in obesity development. Unfortunately, I could not grasp valuable and organized information from the following table 1 to 3 and figure 3, which just showed a bunch of proteins, small molecules, inhibitors, and inducers related to GPR78. Since this part is the principal value of the current review, these messages should be classified and organized, and further, the authors, based on this information, give the optimal interpretation and viewpoints rather than deliver a bunch of messages that seem to have no regulation. This way, the IPA analysis could provide this service and even interpret the mutual relationship or tree map.
5) Perspective, section 5, the comments seem unrelated to the descriptions in the previous paragraph in the current review. Moreover, it only delineated that warrants an exciting direction for future research. Therefore, I suggest the authors conclude the previous paragraphs by raising your comments and perspectives on therapy and drug development for obesity through the GPR78 mechanism.
6) Besides, I suggest the authors add some paragraphs showing GPR78-related gene therapy, which is also a new trend.
Author Response
- The word size in the different paragraphs is not standardized.
R: Changes have been made throughout the entire manuscript.
- Sections 2 to 4.5 explain the function, molecular mechanism, physiological role, role related to disease, pathological function…etc. Therefore, it is suitable for this review to initiate and emphasize the role of GPR78 in obesity and the value used for the drug development target. However, these kinds of reviews are too many. The authors can find many publications only delineate the physiological and pathological functions of GPR78. So, the value of this review explores the role of GPR78 as a critical modulation factor for the treatment of obesity and is the most crucial paragraph in section 4.6. However, section 4.6 delineates the protein and small molecules that directly bind with GPR78 and the molecular mechanism of GPR78 in obesity development. Unfortunately, I could not grasp valuable and organized information from the following table 1 to 3 and figure 3, which just showed a bunch of proteins, small molecules, inhibitors, and inducers related to GPR78. Since this part is the principal value of the current review, these messages should be classified and organized, and further, the authors, based on this information, give the optimal interpretation and viewpoints rather than deliver a bunch of messages that seem to have no regulation. This way, the IPA analysis could provide this service and even interpret the mutual relationship or tree map.
R: Changes have been made as recommended. IPA analysis is now added. Section 4.6 is changed to section 5, and is divided into 2 subsections: “5.1 Proteins and small molecules directly bind GRP78” and “5.2 GRP78 monomer/heteromer and conformational changes”. Mainly explain what kind of molecular improve obesity via directly targeting on GRP78 (Figure 3, Table 3), and GRP78 functions as obesity target, what might be its downstream direct acting proteins (Figure 1, Table 1). Target engagement including direct acting proteins, molecules and downstream signal pathway are the core of this section 5. Different from section 5.1, another allosteric regulation of GRP78 is discussed in section 5.2, which also is a hot topic of current research. As described in section 5.1 and 5.2, four hypotheses of GRP78 activity moderation are summarized: GRP78 transgene expression, ATPase activity, secretion, and ubiquitin-mediated degradation.
- Perspective, section 5, the comments seem unrelated to the descriptions in the previous paragraph in the current review. Moreover, it only delineated that warrants an exciting direction for future research. Therefore, I suggest the authors conclude the previous paragraphs by raising your comments and perspectives on therapy and drug development for obesity through the GPR78 mechanism.
R: Changes have been made as recommended. We conclude the previous paragraphs and raising comments and perspectives
- Besides, I suggest the authors add some paragraphs showing GPR78-related gene therapy, which is also a new trend.
R: Changes have been made as recommended. GPR78-related gene therapy mentioned in Table 2 (RGD ligand-directed phage with GRP78 promoter; RGD4C/AAV/phage binding activates therapeutic transgenes expression with GRP78 promoter). Gene therapy is out of the scope of this review.
Round 2
Reviewer 2 Report
The authors replied to almost all of my questions and added the analysis I suggested in the previous review. Currently, this manuscript is acceptable for publication.